# Quantification and Visualization of Reliable Hemodynamics Evaluation Based on Non-Contact Arteriovenous Fistula Measurement

**DOI:** 10.3390/s22072745

**Published:** 2022-04-02

**Authors:** Rumi Iwai, Takunori Shimazaki, Yoshifumi Kawakubo, Kei Fukami, Shingo Ata, Takeshi Yokoyama, Takashi Hitosugi, Aki Otsuka, Hiroyuki Hayashi, Masanobu Tsurumoto, Reiko Yokoyama, Tetsuya Yoshida, Shinya Hirono, Daisuke Anzai

**Affiliations:** 1Faculty of Clinical Engineering, Hyogo Medical University Hospital, Nishinomiya 663-8501, Japan; 2Department of Clinical Engineering, Faculty of Health Care, Jikei University of Health Care Sciences, Osaka 532-0003, Japan; t-shimazaki@juhs.ac.jp (T.S.); y-kawakubo@juhs.ac.jp (Y.K.); 3Graduate School of Engineering, Osaka City University, Osaka 558-8585, Japan; ata@osaka-cu.ac.jp; 4Department of Medicine, School of Medicine, Kurume University, Fukuoka 830-0011, Japan; fukami@med.kurume-u.ac.jp; 5Department of Dental Anesthesiology, Faculty of Dental Science, Kyushu University, Fukuoka 812-8582, Japan; yokoyama@dent.kyushu-u.ac.jp (T.Y.); hitosugi.takashi.724@m.kyushu-u.ac.jp (T.H.); 6Hemodialysis Center, Sugi Hospital, Fukuoka 837-0916, Japan; ohtsukaki@sugi-hosp.jp; 7Hemodialysis Center, Tamaki-Aozora Hospital, Tokushima 779-3125, Japan; h.hayashi@tamaki-aozora.ne.jp; 8Department of Clinical Engineering, Faculty of Health and Welfare, Tokushima Bunri University, Kagawa 769-2193, Japan; tsurumoto@kgw.bunri-u.ac.jp; 9Department of Clinical Laboratory and Laboratory Medicine, Kyushu University Hospital, Fukuoka 812-8582, Japan; yokoyama.reiko.272@m.kyushu-u.ac.jp; 10Foundation for Biomedical Research and Innovation at Kobe, Kobe 650-0047, Japan; t-yoshida@fbri.org; 11Hemodialysis Center, Wakaura Central Hospital, Wakayama 641-0054, Japan; touseki@wachuohp.or.jp; 12Graduate School of Engineering, Nagoya Institute of Technology, Nagoya 466-8555, Japan; anzai@nitech.ac.jp

**Keywords:** hemodialysis, hemodynamics, arteriovenous fistula, non-contact measurement

## Abstract

The condition of arteriovenous fistula (AVF) blood flow is typically checked by using auscultation; however, auscultation should require a qualitative judgment dependent on the skills of doctors, and further attention to contact infection is required. For these reasons, this study developed a non-contact and non-invasive medical device to measure the pulse wave of AVFs by applying optical imaging technology. As a first step toward realization of the quantification judgment based on non-contact AVF measurement, we experimentally validated the developed system, whereby the hemodynamics of 168 subjects were visually and quantitatively evaluated based on clinical tests. Based on the evaluation results, the fundamental statistical characteristics of the non-contact measurement, including the average and median values, and distribution of measured signal-to-noise power ratio, were demonstrated. The clinical test results contributed to the future construction of quantified criteria for the AVF condition with the non-contact measurement.

## 1. Introduction

Currently, ~3.5 million patients are receiving hemodialysis therapy worldwide [1]. In Japan, ~340,000 dialysis patients were treated in 2019 [2], which implies that 1 in 380 people in Japan is receiving treatment. Hemodialysis therapy is one of the most common methods for blood purification, in which blood is pumped out of the body and passed through a purifier, known as a dialyzer, to remove urinary toxins, regulate electrolyte balance, and exclude excess fluid (Figure 1). The basic cycle of the treatment is 4 h per session, with 3 sessions per week. To perform the hemodialysis therapy, a unique blood vessel, known as the arteriovenous fistula (AVF), is used. This vessel is surgically created by anastomosing the flexural cutaneous vein and artery in distal and radial positions. As the intravascular pressure is higher in the radial artery than in the flexor cutaneous vein and the central resistance is lower than the peripheral vascular resistance at the fingertip, the arterial blood flows into the flexor cutaneous vein. These phenomena cause turgid veins and hyperreflux, which generate in an access line for blood delivery and withdrawal.

As the venous wall is thinner and softer than the arterial wall [3], the turbulent flow at the anastomotic site causes the venous walls to collide with each other. As a result, a fistula bruit sound known as a vascular murmur and a vibration known as a thrill [4] occur. The thrill can be measured by lightly touching the anastomosis with a finger. The AVF pulse wave $W_{\mathrm{AVF}}\left(t\right)$ can be decomposed as
(1)$$W_{\mathrm{AVF}}\left(t\right)=W_{heartbeat}\left(t\right)+W_{thrill}\left(t\right)$$
where $W_{heartbeat}\left(t\right)$ and $W_{thrill}\left(t\right)$ are the pulse wave components of the heartbeat and thrill, respectively.

As the AVF often becomes stenotic [5], the AVF condition is checked by auscultation before and after thedialysis therapy, especially in Japan [6]. Auscultation is an excellent initial screening method owing to its simplicity and reliability. However, it can only help make qualitative decisions. In other words, the auscultation method confirms two kinds of AVF states: normal or abnormal condition. There are other issues as well. For exmple, auscultation results are affected by the pressure applied to the AVF vessel, it occasionally misses the sound of stenosis, the obtained data cannot be shared with co-workers, even if the co-workers are in the same hospital, and the decision can only be made by the staff who perform the measurement, and careful attention to contact infection is necessary because of the direct contact required with the patient.

To solve these problems, existing studies [7,8,9,10,11,12,13,14] developed a system with an electronic stethoscope to detect a fistula bruit (shunt sound) by using wavelets [7,8,9,10,11,12] and deep learning [13,14]. However, electronic stethoscope-based detection is sensitive to environmental noise [15]. In other related studies [15,16,17], the AVF pulse wave acquired by a photoplethysmogram was analyzed based on wavelet [15] and neural network analysis [16,17]. The photoplethysmogram still depends on the amount of pressure exerted by the sensor on the skin while in contact. Kamiyama et al. attempted to detect the stenosis site by capturing its image from above by using a charge-coupled device (CCD) camera [18]. Zhu et al. estimated the blood flow amount based on the weak AVF movement by capturing the image from a side view by using a digital camera [19].

Optical technology, e.g., photographic imaging based on a CCD camera, is efficient in realizing the non-contact and non-invasive measurement of the AVF condition. The advantage of the non-contact evaluation is to avoid concerns about infections because dialysis patients in particular are often infected with hepatitis viruses. In addition, contact AVF measurement should depend on the contact condition, so the non-contact measurement can be expected to solve the problem of the contact condition. However, in existing studies, the validation was limited because the phantom evaluation was conducted [18], and clinical tests with a sufficient number of subjects were not carried out [19]. Furthermore, the quantification of the AVF hemodynamic condition was rarely discussed, which is required to share the diagnostic results. Therefore, this study aims to develop a non-contact device to quantitatively and visually measure the AVF hemodynamic conditions by applying optical technology with color mapping. To validate the developed system, we conducted clinical experiments to clarify hemodynamics in 168 subjects. Then, a moving-average filter to extract AVF pulse waves with high signal-to-noise power ratios (SNRs) is required to quantify the AVF condition. In addition, this paper demonstrates the effectiveness of hemodynamic visualization and quantification based on the clinical test results, which implies that AVF quantification can solve the problems of the conventional qualitative diagnosis and significantly improve hemodialysis therapy.

## 2. Principle of Non-Contact AVF Pulse Wave Measurement

### Moving-Average Filter

Let us explain the principle of the quantification of the AVF pulse wave. We apply moving-average filtering to the measured image data. In the moving-average filter method, the kernel is defined as an N×M matrix whose component represents a weight coefficient h(m,n). The original image data f(i,j) are convolutionally integrated with the kernel while moving one pixel at a time in a raster scan. Thus, we obtain the moving-averaged value g(i,j) as
(2)$$g\left(i,j\right)=\sum_{n=-1}^{N}\sum_{m=-1}^{M}f\left(i+m,j+n\right)h\left(m+1,n+1\right).$$

The purpose of moving-average filtering is to suppress scattered reflections and halos in the original image data. As the imaging data include pulse wave data for each pixel, a large amount of pulse wave data can be obtained at the same time. Therefore, an increase in the kernel exhibits an effect similar to that of additive averaging. The additive averaging method is effective in extracting weak biological signals from the superimposed noise. Similar techniques were applied to evoke potential measurements [20,21].

## 3. Methods

### 3.1. Experimental Protocol

In the clinical test, 168 subjects were rested in a seating position and imaged twice at 40 fps (frames per second) for 10 s at two points close to the apex of the anastomosis by using the developed non-contact imaging system (Figure 2). Table 1 and Table 2 show information and statics of the patients. The file format obtained from the AVF imaging was a raw image format with RGB information for each pixel. Figure 3 shows the setup and analysis flow of the non-contact AVF measurement system. In accordance with the Declaration of Helsinki, sufficient attention was paid to the protection of all subjects. The approval was obtained from the ethics committee of the hospital. The hardware and software specifications of the non-contact imaging system are listed in Table 3.

### 3.2. Hemodynamics Visualization Method Based on Color Mapping

The visualization of hemodynamics helps evaluate the AVF condition. In this study, color-mapping technology has been applied for visualization, wherein the blood flow direction, which is ejected from the anastomosis toward the trunk side, can be confirmed. Specifically, hemodynamics are visualized by using time-series data in which changes in luminance of the pixels of the raw image format are converted to a 256-level color tone (Figure 4).

### 3.3. Determination of the Optimal Kernel Size and Quantification

The AFV pulse wave waveform was obtained by increasing the kernel size of the moving average filter from 1 pixel (0.08 nm) to 201 pixels (16.08 nm) at every interval of 2 pixels from a 1 × 1 matrix to a 201 × 201 matrix. Figure 5 shows the relationship between the AVF imaging area and kernel size of the moving-average filter.

Next, let us explain the quantification of hemodynamics for AVF evaluation. The quantification based on the obtained image data is performed by using the following procedure. Because the typical heart rate is between 60 bpm and 100 bpm [22], the lower and higher cutoff frequencies of the band-pass filter (BPF) to extract the pulse wave component is set to 1 Hz and 5 Hz, respectively, based on our previous study [23]. Moreover, the lower and higher cutoff frequencies of the BPF to extract the noise component are set to 10 Hz and 19 Hz, respectively, so that the thrill wave frequency band can be ignored. A higher cutoff frequency of 19 Hz is set because of the Nyquist frequency. The obtained waveforms have been processed by using a fast Fourier transform with a sample size of 256 to realize each BPF to divide the waveform into pulse wave and noise components. Then, we have calculated the SNR as the ratio of the signal (pulse wave) and noise powers.

To calculate the SNR, the method to determine signal and noise components should be discussed. In this study, as noise may be mixed with the AVF component, the sum of the powers at the fundamental frequency and second harmonic components is used as the pulse wave domain band (Figure 6). In addition, the sum of the first- and second-largest signals at 10–19 Hz is considered as the noise domain. The optimal kernel size is determined to achieve the highest SNR; therefore, the quantification of the AVF condition can be defined as:(3)$$SN_{\mathrm{AVF}}=\max\left({\mathrm{AVF}}_{s}/{\mathrm{AVF}}_{n}\right)$$
where ${\mathrm{AVF}}_{s}$ and ${\mathrm{AVF}}_{n}$ denote the powers of the pulse wave and noise components, respectively.

## 4. Results

### 4.1. Preliminary Evaluation: Effect of Changes in Luminance, Hue, and Color Saturation on Pulse Waveform

Next, we investigated the optimal measure based on the AVF imaging data. Figure 7 shows a comparison among the observed pulse wave waveforms with respect to changes in luminance, hue, and saturation [24] by irradiating white light on the palms of a healthy subject. The evaluation measure of "luminance” is the luminous flux of a surface light source when viewed from an observation point that is expressed in $\mathrm{cd}/m^{2}$. The word "hue” defines the characterization of a color, and "saturation” is the degree of vividness of a color. The measure of "lightness” is an expression similar to luminance; however, the measure of lightness’ is excluded in the evaluation process because it is a psychophysical quantity related to vision that is recognized by reflected light.

Based on the comparison results, as shown in Figure 7, it is confirmed that the pulse pressure is calculated based on the luminance variation (measured by the peak-to-peak value of the waveform), which demonstrated the largest value among the three results, and the superimposed noise was relatively small. Hence, we decided to adopt the luminance variation of the AVF pulse wave measurement in the developed non-contact AVF measurement system.

### 4.2. Preliminary Evaluation: Optimal Wavelength of Imaging

In commercially available watch-type healthcare terminals, as well as in our previous studies, green light is an effective light source for contact measurement methods [25,26,27,28]. However, for non-contact AVF pulse wave imaging, the wavelength for reliable imaging has not yet been clarified. To determine the optimal wavelength for non-contact AVF imaging, it is necessary to conduct preliminary experiments by using several wavelengths, i.e., green light and blue light.

Figure 8 shows the measurement of the pulse wave waveform of the anastomotic area with blue light. In the experiments, effective waveforms were obtained with green land blue lights. However, as a result of three-dimensional image processing of the artery using a photometric method, it became difficult to observe three-dimensional imaging with green light. This is because the photometric method [29,30] is a technique to synthesize three-dimensional images from planar images captured from different directions. As the wavelength of the photometric method is sufficient accurate for small veins on the surface layer, it is difficult to create three-dimensional images owing to the longer wavelength compared to that of blue light. Because this phenomenon is unique to non-contact AVF imaging, we decided to use blue light as the source.

### 4.3. Preliminary Evaluation: Validation of Non-Contact AVF Imaging

Finally, to validate the pulse wave measured by the non-contact AVF imaging device, we confirmed how a large correlation between the radial artery waveforms and the developed non-contact imaging of a bidirectional Doppler flowmeter (ES-100V3, Hadeco, medical device certification number 21500BZZ00131000). Note that this Doppler flowmeter is based on contact measurement. We carried out the measurement with one healthy subject as shown in Figure 9. As shown in Figure 10, the developed non-contact system ensures the peak period corresponds to the commercial medical device with contact measurement. In other words, the blood flow meter matched the systolic cycle of the heart with regard to the measurement data. Based on the preliminary evaluations, we confirmed that the developed non-contact system has a fundamental function to measure the systolic cycle of the heart.

### 4.4. Color Mapping-Based Hemodynamic Visualization

Figure 11 shows the continuous AVF imaging results for one cycle (from 2.600 s to 3.360 s after the start of imaging) of a subject. At 2.850 s, the AVF was indicated in blue, and the anastomosis was tense owing to ejection. At 3.000 s, the blood moved from the anastomosis to the puncture site, and at 3.125 s, the puncture site was represented at its peak. At 3.150 s, the entire forearm turned blue. Finally, at 3.250 s, the intravascular pressure drastically dropped, and the anastomosis and puncture site changed in a reddish cycle, which was observed in other subjects as well.

### 4.5. Measurement Data with Moving-Average Filtering

Figure 12 shows the AVF pulse waves for the kernel size of the moving-average filter *N* = 1, 13, and 25. Each pulse wave waveform was normalized for comparison. It was confirmed that the AVF pulse wave was clearly obtained, suppressing the noise component, as the kernel size increased.

### 4.6. AVF Quantification with Optimal Kernel Size

As shown on the left side of Figure 13, the SNR curve is close to an exponential function as *N* increases. The highest $SN_{\mathrm{AVF}}$ = 0.18 was obtained when the kernel size N=113 was the optimal value. The normalized distribution of $SN_{\mathrm{AVF}}$ with N=113 is shown on the right side of Figure 13. The median $SN_{\mathrm{AVF}}$ value of the normalized distribution was −0.37. Additionally, the validation results demonstrated that the first quartile (25 %tile), second (50 %tile), third (75 %tile), and the fourth quartiles (the maximum value) were −0.77, −0.60, 0.23, and 1.44, respectively.

Note that the quartile can be defined as a type of quartile that divides the number of measurement data into four parts with an equal size. For example, the first quartile (25 %tile) is calculated as the middle number between the minimum and the median of the measurement dataset. Additionally, to calculate the normalized distribution of $SN_{\mathrm{AVF}}$, we balanced the measurement data by the average value. Therefore, the average value (not median value) of normalized $SN_{\mathrm{AVF}}$ should be of 0 and span from negative to positive value. We can see from Figure 13 that the median value (2nd quartile) is not different from the average value. Also, the variation of normalized $SN_{\mathrm{AVF}}$ ranges from −0.77 to 1.44. It is difficult to construct certain criteria to judge the AVF condition based on only the results; however, the results can be used to realize the quantified evaluation of AVF measurement. For example, this study obtained the $SN_{\mathrm{AVF}}$ distribution from the experiment with 168 normal dialysis patients in this study, so these data should be informative to analyze the experiment with non-normal patients in the future.

Furthermore, Figure 14 shows typical examples of the AVF pulse wave measurement results with optimal kernel N=113. As can be seen from these results, the AVF pulse wave waveforms were efficiently measured in the developed system. In addition, for providing the statics of the measured data, Figure 15 demonstrates the average and variance of the measured luminance for all patients.

### 4.7. Observation of the Reversed-Phase Pulse Wave

Although no reversed-phase signal component is observed in a normal pulse waveform, a reversed-phase waveform, as shown in Figure 16, was observed at the AVF periphery for all the subjects.

## 5. Discussion

### 5.1. Visualization by Color Mapping

Here, we discuss the reason for the reversed-phase waveform observed in the AVF pulse wave (Figure 16). To explain this, Figure 17 shows the color-mapping results of the AVF pulse waves of the parietal (01) and lateral (02) portions. When the arterial blood is ejected and the intra-anastomotic pressure is maximum, the CCD camera and AVF become closest to each other. Hence, the light received at the parietal part exhibits the largest amount. At point (01) on the parietal, the light-receiving volume reaches its maximum value at 0.4 s; however, the light-receiving volume decreases as the intravascular pressure decreases and reaches its minimum value at 0.9 s. If the principle of waveform acquisition is based on the same absorbance change as in general photoplethysmographs, the minimum value can be obtained at 0.4 s at the top of the head and the maximum value at 0.9 s. For this reason, it can be considered that the non-contact AVF pulse wave is a displacement change caused by the vibration of the CCD camera and AVF.

However, the side (02) is a reversed-phase waveform with a minimum value at 0.4 s because the shadow becomes larger as the AVF increases. The maximum value at 0.9 s is observed because the light-receiving amount increases again as the AVF decreases. This phenomenon can be explained by the alternation of the positive reflection of the parietal surface and the shadow of the lateral surface of the light caused by the vibration of the blood vessel. This is why the present method is based on the displacement change rather than the absorbance change. Because this phenomenon does not occur in the contact method, which is unique to non-contact, it should be considered while measuring large vessels that produce shadows in a non-contact manner.

The reason for the increase in luminance over the entire forearm at 3.150 s, as shown in Figure 11, can be explained as follows: as shown in Figure 18, the blood driven from the heart flows through the brachial artery and branches into the radial and ulnar arteries to the peripheral vessel. The blood returns via the cephalic vein to the heart. Thus, in addition to the normal pathway through the peripheral vessel, the bloodstream of the cephalic vein has two pathways that provide a short route toward the peripheral vessel.

As this phenomenon occurred during the anastomotic relaxation phase, normal blood flow was considered through the peripheral vessels. This allows us to identify the site of stenosis not only in the anastomotic area but also in the entire forearm. The visualization of the blood flow from the AVF back to the trunk side based on color mapping can be used to identify the site of stenosis. In this study, the quantization of the color mapping was 8 bits; however, we assume that higher resolution color mapping will be possible by improving quantization.

### 5.2. AVF Pulse Wave and Stereoscopic Image with Blue Light

To determine the wavelengths in the preliminary experiments, blue and green lights were considered in the AVF pulse wave measurement. In the preliminary experiment, the pulse waves were observed with almost no inferiority in the cases of green and blue lights. This is because while general photoplethysmography (PPG) uses the absorbance change associated with arterial pulsation, this method is based on the displacement change between the camera and AVF associated with the pulsation of the AVF. Furthermore, the reason why we could not create a three-dimensional image with green light is that the anastomosis is a large blood vessel; therefore, it absorbs a large amount of light, and the reflected light decreases. On the other hand, as most of the blue light is reflected by the epidermis, the reflected light increases, and a three-dimensional image can be created.

### 5.3. Effect of Moving-Average Filtering

Finally, this paper discusses the reason why a curve exhibits a trend that a sharp increase in SNR before the peak of 0.179 (N=113) and gradual decrease after the peak (Figure 13). The pulse wave component can be assumed to be a periodic signal obtained from each pixel in the same phase. Moreover, the noise component can be assumed to be random noise caused by scattered reflection obtained from each pixel, where we can confirm the effect of moving-average filtering [31]. Figure 19 shows the left figure of Figure 13 in an expanded view with N= 65–201. The SNR slowly decreases from N=113, and sharply drops from N=187. As *N* increases, the waveform at the side of the anastomosis is averaged, and the reverse phase waveform is added to the average. In this study, the bandwidth of the AVF observation was set to 1–5 Hz; however, a few thrill waveforms were observed in this band; therefore, it is suggested that thrill waveforms exist in the frequency band of 5–10 Hz, which was excluded in this study. Therefore, the frequency range of the thrill waveforms can be used as a substitute for palpation.

### 5.4. Limitation of Developed Non-Contact AVF Evaluation System

Here, let us discuss the limitation of the developed system. In the developed system, the test results were strongly affected by ambient light because the detection of the luminance should include not only the change of blood pulse but also the ambient light. Hence, the developed system employs a light-shielding box to prevent the ambient light from being superimposed on the measurement data.

It is noted that, as for the authorization for the present method, careful discussion is needed to obtain CE marking or any market; however, the developed system does not require strict selection criteria for the patient or any special skills of doctors, so that, it can be widely spread to the market.

## 6. Conclusions

This study has developed a non-contact AVF measurement system for the visualization and quantification of hemodynamics in order to solve the problems of contact auscultation. In the preliminary experiments, luminance has been found to be a better choice to acquire the AVF pulse wave waveform than hue and saturation. Furthermore, in existing studies of AVF measurement based on optical technology, validation was rarely conducted based on clinical tests with a sufficient number of subjects. Therefore, this study has conducted clinical tests with 168 normal dialysis patients to validate the quantification of non-contact AVF measurements. From the clinical test results, the median SNRAVF value of the normalized distribution was −0.37, and the first (25 %tile), second (50 %tile), third (75 %tile), and fourth quartiles (the maximum value) were −0.77, −0.60, 0.23, and 1.44, respectively, which should be informative to construct criteria for the quantified evaluation of AVF measurement.

Currently, we are carrying out another clinical test to make measurements before and after stenosis to determine the hemodynamics of the stenosis site to realize early detection of abnormalities. In the future, we will apply and use the developed system to the AVF condition evaluation and for patients with peripheral circulatory failure, respectively.

## Figures and Tables

**Figure 1 sensors-22-02745-f001:**
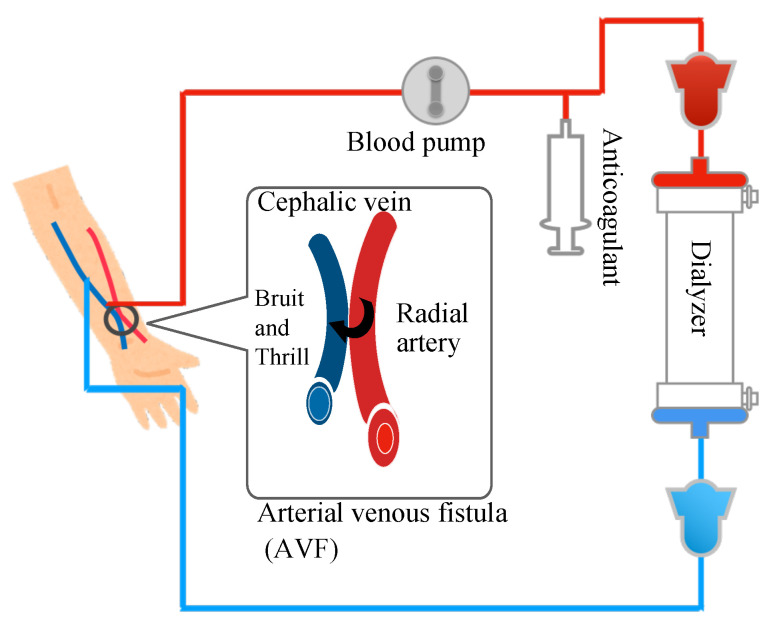
Overview of hemodialsis therapy.

**Figure 2 sensors-22-02745-f002:**
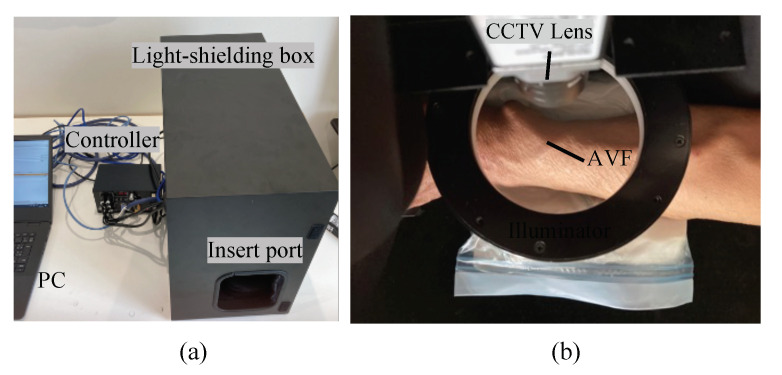
Experimental equipment. (**a**) External view of the experimental equipment. (**b**) Internal view of the light-shielding box.

**Figure 3 sensors-22-02745-f003:**
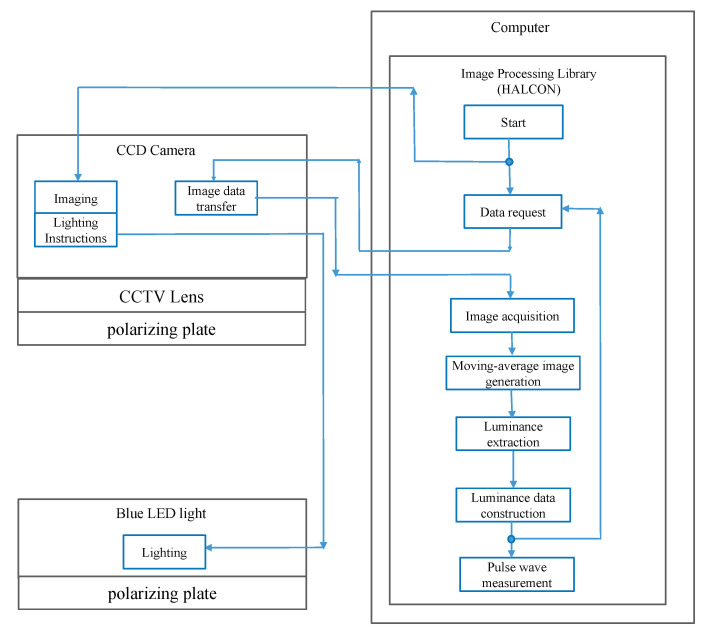
Setup and analysis flow of the non-contact AVF measurement system.

**Figure 4 sensors-22-02745-f004:**
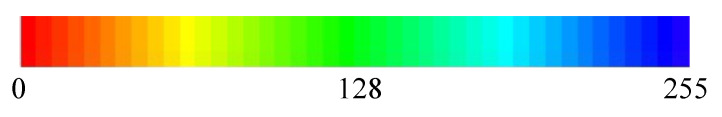
Correspondence between brightness and color tone for the color mapping.

**Figure 5 sensors-22-02745-f005:**
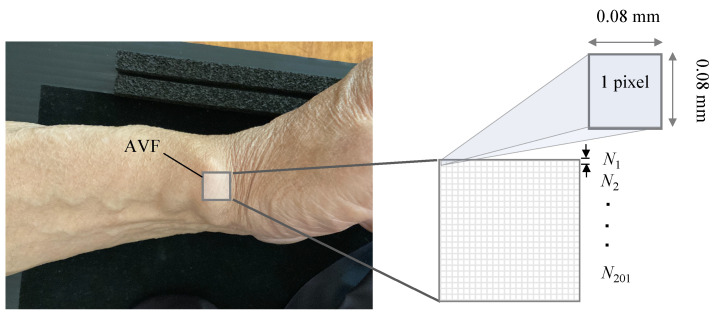
AVF imaging area and the kernel size (N×N) of the moving-average filter.

**Figure 6 sensors-22-02745-f006:**
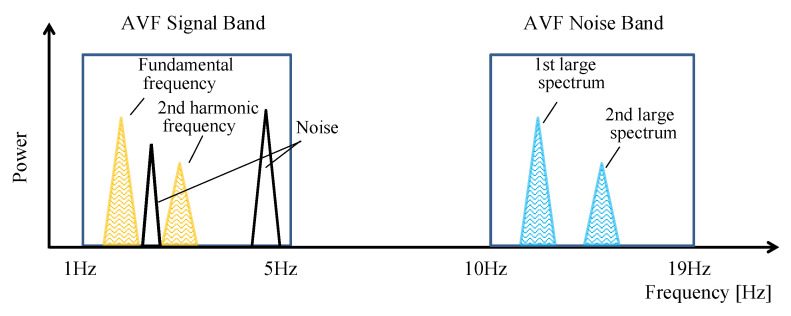
Frequency spectrum of AVF pulse wave waveform.

**Figure 7 sensors-22-02745-f007:**
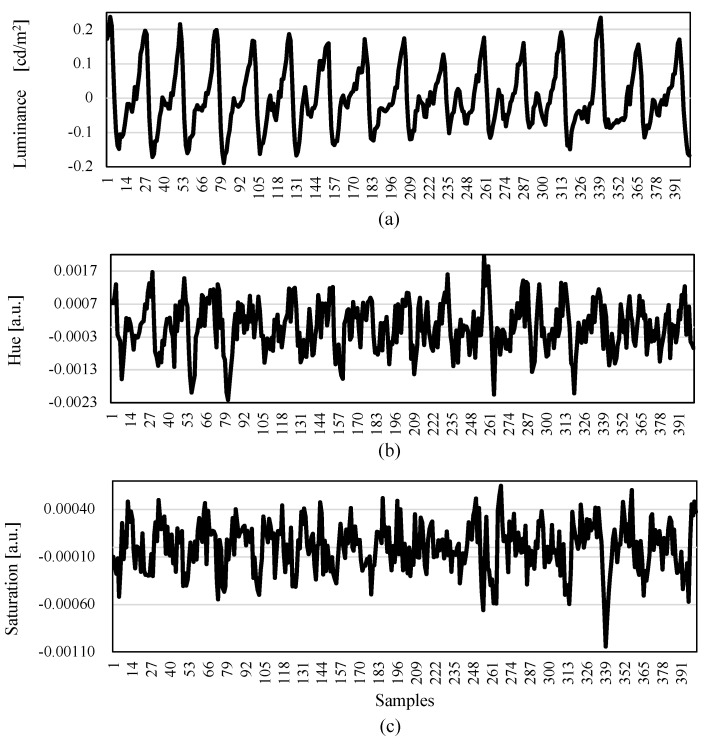
Pulse wave measurement based on changes in (**a**) luminance, (**b**) hue, and (**c**) saturation. The measurement was conducted on the palm of a healthy person’s hand. The horizontal axis shows the data sample index with time resolution of 25 ms (the sampling frequency of 40 Hz). It was confirmed that the best way to measure the AVF pulse wave was achieved based on the change with the luminance.

**Figure 8 sensors-22-02745-f008:**
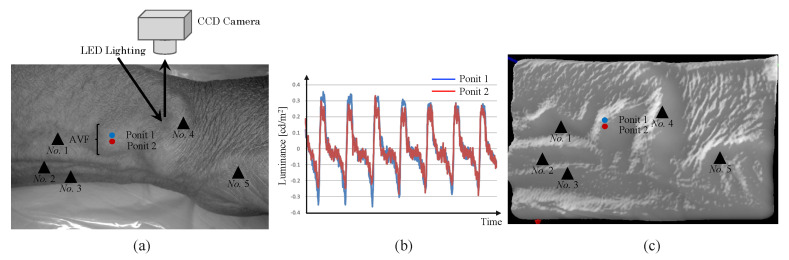
AVF pulse wave measured with blue light. (**a**) Photograph of a left wrist. (**b**) AVF pulse wave measured at the two points close to the anastomosis. The low- and high-cutoff frequencies of band pass filter were 1 Hz and 10 Hz, respectively. Because the similar waveforms are observed at different points, the measured data is not sensitive to observation points. (**c**) Three-dimensional image by the photometric method with blue light. As compared with the photo shown in (**a**), good agreement is confirmed, which means that the three-dimensional image based on the luminance is efficient to represent the AVF form.

**Figure 9 sensors-22-02745-f009:**
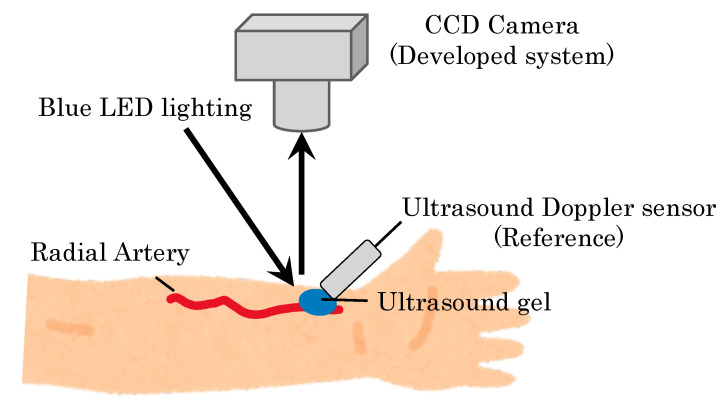
Setup for comparison between contact and non-contact measurements.

**Figure 10 sensors-22-02745-f010:**
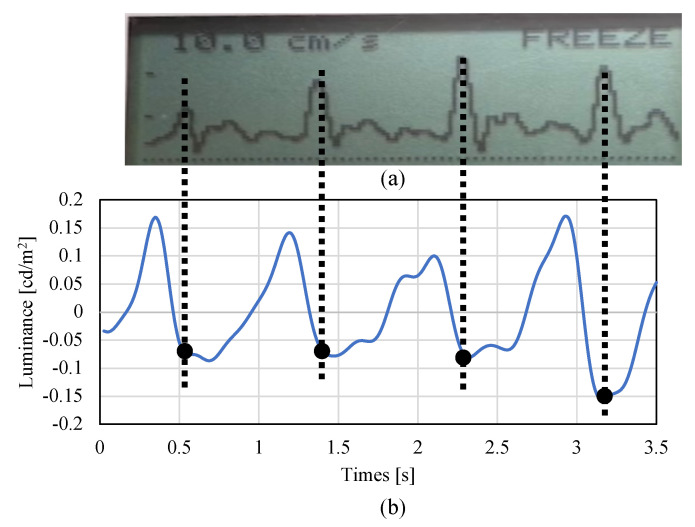
Comparison with reference data. (**a**) Reference data obtained by bidirectional Doppler flowmeter. (**b**) Measurement data by the developed system with blue light source at radial artery. Since the observed pulse wave was inverted in the developed system, both periods were compared in the minimum values. It was confirmed that the period of the peak value of the blood flow meter (reference) and the peak value of the radial artery were in agreement.

**Figure 11 sensors-22-02745-f011:**
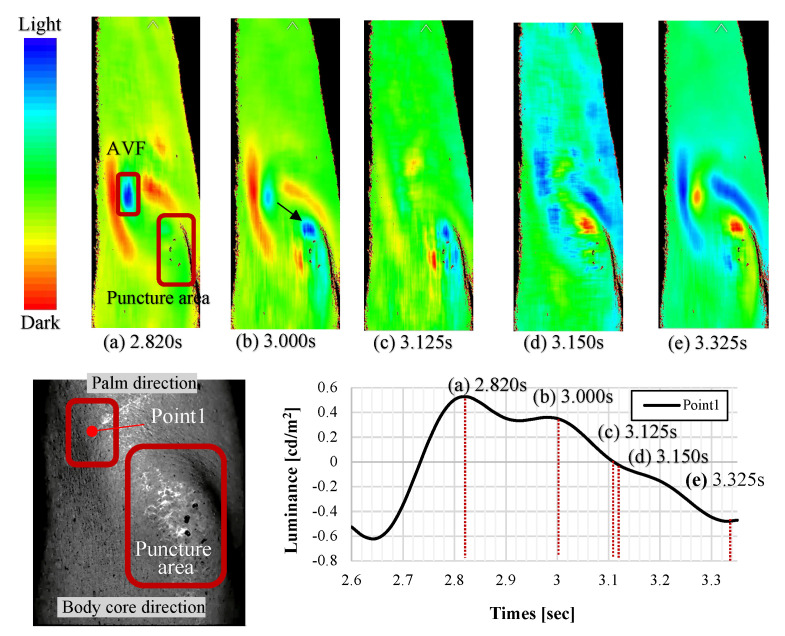
Continuous color -mapping image of AVF pulse wave for one cycle (2.600 to 3.360 s, the band pass filter was applied). (**a**) Anastomotic tense phase (2.820 s). (**b**) Transition from anastomosis to puncture site (3.000 s). (**c**) Peak puncture site phase (3.125 s). (**d**) Anastomotic relaxation phase (3.150 s). (**e**) End of anastomotic relaxation phase (3.325 s).

**Figure 12 sensors-22-02745-f012:**
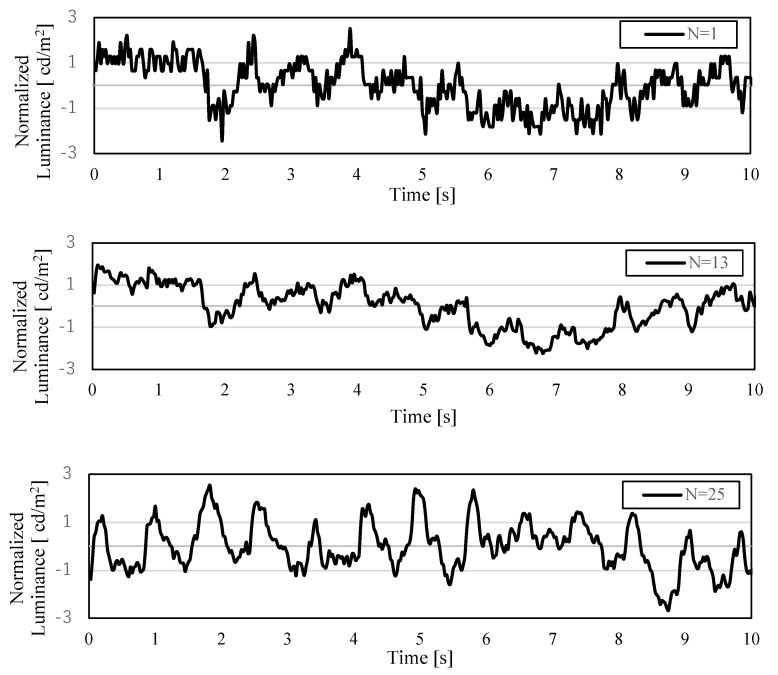
Effect of the kernel size *N* on noise reduction in the AVF pulse wave measurement. The random noise on the AVF pulse wave was suppressed by the moving-average filtering. Larger kernel size made the observed AVF pulse wave clearer.

**Figure 13 sensors-22-02745-f013:**
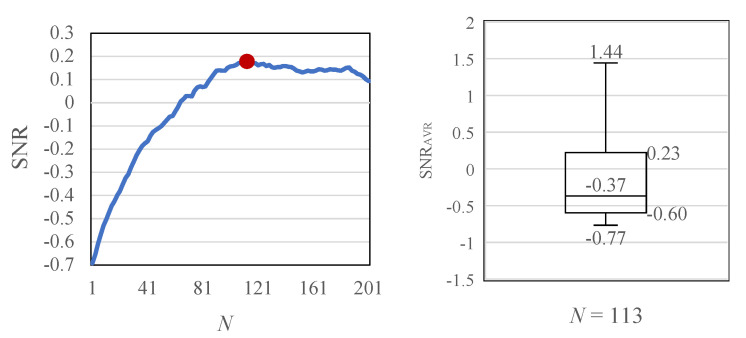
Determination of optimal kernel size *N*. $SN_{AVF}=0.179$ was achieved when the optimal kernel N=113.

**Figure 14 sensors-22-02745-f014:**
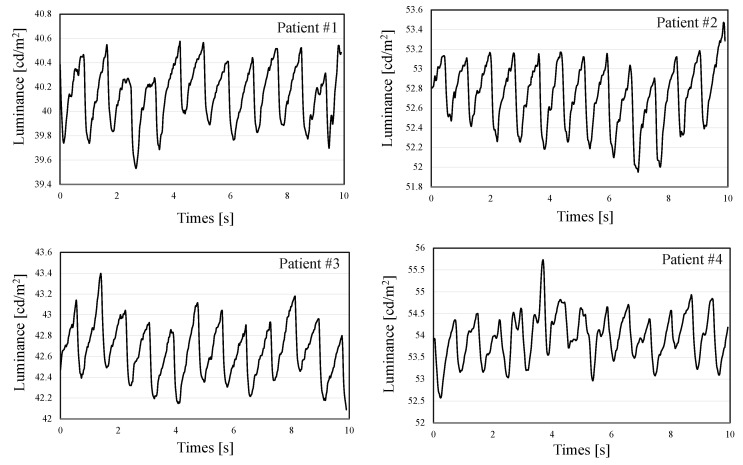
Typical examples of AVF pulse wave measurement with optimal kernel N=113.

**Figure 15 sensors-22-02745-f015:**
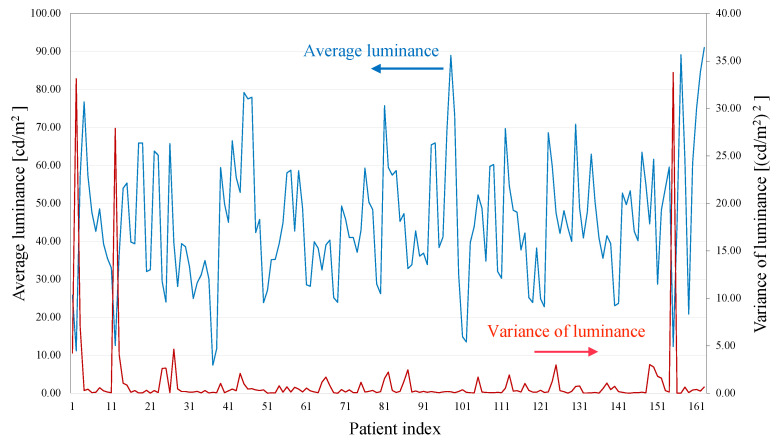
Average and variance of measured luminance for all patiants with optimal kernel N=113.

**Figure 16 sensors-22-02745-f016:**
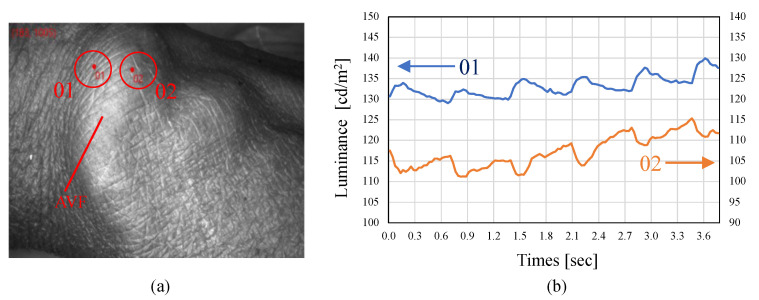
Measurement example of reversed-phase AVF pulse wave. (**a**) Photograph of measurement points on AVF. (**b**) Measured waveforms obtained at the two measurement points.

**Figure 17 sensors-22-02745-f017:**
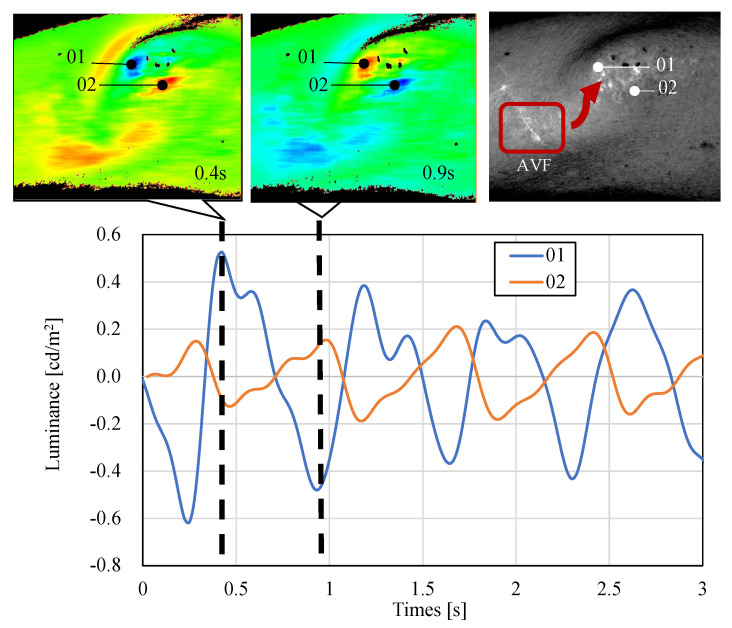
Detailed demonstration of the reversed-phase AVF pulse waveform. The luminance of the (01) measurement point at 0.4 s increased, whereas that of the (02) point decreased. On the other hand, at 0.9 s, the luminance of the (02) point increased, and that of the (01) point inversely decreased. It was suggested that the shadows caused by the masses may have an effect on the phenomenon of the reversed-phase AVF pulse waveform.

**Figure 18 sensors-22-02745-f018:**
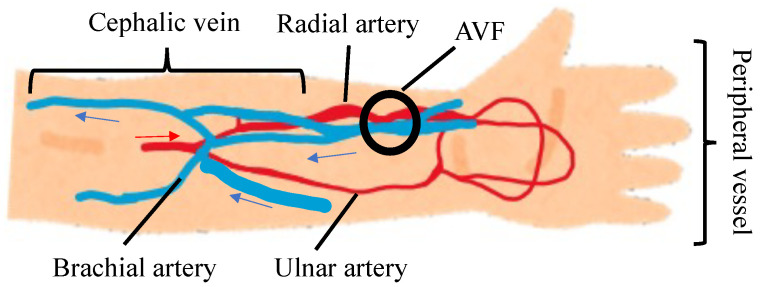
Mechanism of blood flow in cephalic vein.

**Figure 19 sensors-22-02745-f019:**
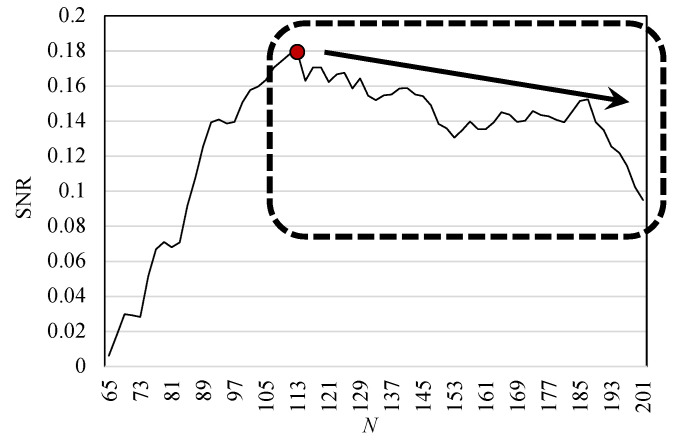
Expanded view of the SNR variation with the kernel size *N* = 65–201. The reason for the slow decrease in SNR from $SN_{\mathrm{AVF}}=0.179$ (N=113) can be explained by the fact that the reversed-phase waveform at the side of the anastomosis is also averaged as the kernel size *N* increases.

**Table 1 sensors-22-02745-t001:** Information and protocol of clinic test.

Date	From 12 October to 5 November 2021
Place	Tamaki-Aozora Hospital, Tokushima, Japan
Number of patients	168
Patient selection criteria	Patient who has a normal shunt condition with a blood flow of more than 200 mL/min
Preparation	Purpose and rule of the test were sufficiently explained to all patients beforehand
Rules for rejection of tests	They can refuse to participate at any time during the test
Measurement protocol	Measurement in resting sitting condition for 10 s

**Table 2 sensors-22-02745-t002:** Statics of patients.

Average age	69.1 years old
Average dialysis years	8.2 years
Average blood flow	226.8 mL/min
Standard deviation of blood flow	28.0 mL/m

**Table 3 sensors-22-02745-t003:** Hardware specifications.

Parts	Model Number	Venders	Features
Image processing software	Halcon Rev.18	MVTec	
CMOS camera	a2A1920–160 μm	BASLER	1920 × 1200 pixels, 160 fps
CCTV lens	FA0802D	CHIOPT	F#1.4–16
Polar screen	#52–556	Edmund Optics	
Illuminator	IMAR–130DB–8ch	LEIMAC	Center wavelength: 465 nm
Polarizer	#45–204	LEIMAC	Mounted in the front of a illuminator and CCTV Lens
Controller	IDGB–30M8PG–TP	LEIMAC	AC 100–240 V, DC 12 V, 30 W

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
