# Peer review of "Quantification and Visualization of Reliable Hemodynamics Evaluation Based on Non-Contact Arteriovenous Fistula Measurement"

_sensors, 2022, doi:10.3390/s22072745_

Round 1
Reviewer 1 Report
The research area is of interest and the topic is actual from the technological and clinical point of view. Despite the manuscript does not provide a high level on novelty concerning sensor design it propose the use of a custom-made equipment for stenosis evaluation which is alternative to standard echography and/or auscultatory methods. However, I found the overall organization of the manuscript poor and sometimes most aspects are overstressed. An overall reorganization of the manuscript is thus needed.
Abstract provides useful information; however, it should be better evidenced why it is important the continuous monitoring of anastomosis and what are the risks of stenosis. On lines L11-14 it is not clear what authors want to evidence by using the parameter signal-to-noise-power ratio and why a value of -0.37 (and others) are important to be evidenced. I would avoid the definition of quartiles.
L25-26 I would evidence that anastomosis can be surgically created in both distal and radial position.
I would merge Fig.1 and 2 since they provide complementary information. In figure 2 I think authors should better specified the names of the waveforms. All the waveforms are blood pulse…also replace “Thill” with “Thrill” in Figure 1 and 2. I’m not sure the sum of the second two waveforms results in the first showed.
In introduction authors evidenced the advantages of non-contact stenosis evaluation. However, it is not clear to this reviewer the importance of having non-contact evaluation using for example complex equipment rather than a contact evaluation using simpler devices.
I found the manuscript difficult to follow for a reader not in the field. In particular I would expect to have the a more depth description of the equipment used and then the signal analysis. Conversely authors start the section 2 introducing the concept of moving average and images characteristics but no information was provided on the setup.
What is reported on X-axis in figure 3? Also, Figure 4 is introduced before it is mentioned in the manuscript and it causes difficulty for the reader. What is the important data showed in figure 4 b? the graph should be improved for example with a legend. Same comment for Figure c. Then in Figure 5 commercial flowmeter is compared with a graph which is difficult to compare since the concept behind that result is not presented so far.
Finally it seems that in section 3 the overall equipment is presented. I suggest so to reorganize the manuscript. Table I should be placed later on in the manuscript.
In figure 7 it is not clear what is dark and what is light.
Section 4 should reports the results of the analysis. However, it seems that most experimental data was previously reported. Authors proposed the use of power SNR for comparison. As evidenced in Figure 12 SNR can span from negative to positive values and it is always less than 1.44. A more detailed discussion of these kind of results is desirable.
A comparison with the actual state of art reviewed by authors in the Introduction should be desirable to have a better overlook on the goodness of the proposed approach.
Conclusion should be summarized and better focused.
Reviewer 2 Report
sensors-1602659-peer-review-v1
Quantification and Visualization for Reliable Hemodynamics Evaluation Based on Non-Contact Arteriovenous Fistula Measurement
The authors investigated new non-contact means to quantify and visualize arteriovenous fistula hemodynamics. The goal of the study is well described and the subject is of certain interest to the reader. This non-contact approach is promising and may be a good alternative to existing methods but the authors fail to provide a demonstration of the method's effectiveness: indeed, the presentation of the test methods and the experimental results are not sufficient to demonstrate the validity of the method (see hereafter the additional comments). The test setup is not well described, the detailed test protocol is not disclosed, measurements are made but not shown, for instance, a Doppler flowmeter is used but no data except a photo of the screen is provided, we do not know how this equipment was used (on one single patient to assess the method or for all patients involved in this trial?). The same comment can be made for the Results section: Authors should provide more experimental data records (data that were already obtained during the trial and that would be ideally shared) together with a careful statistical analysis of the results. To be more convincing, the experimental plan should make possible a direct comparison between the AMF state checked by auscultation (by one or ideally several practitioners) and the one derived from the present method, to show that the non-contact approach allows a reliable and robust determination of this AMF state and finally that the method provides a significant improvement in the AMF state diagnosis.
The limitations of the method should be better discussed: we would like to understand for instance the physical parameters or other conditions that have a positive or negative impact on the test results. Also, the patient’s selection criteria (age, gender, weight…) are not disclosed, it is difficult to understand to which extent the method is reliable or not. Finally, this would be the opportunity to discuss the technical challenges of obtaining CE marking or any marketing authorization for such a medical device.
Additional comments/recommendations:
page 3 line 63: typo
Fig 5 is not very convincing, the authors cannot just scale up a photo to see the good match between the 2 spectra. It is strongly recommended to record both signals using the same DAQ system with a single internal clock that ensures a good synchronization
Page 6 line 121: 168 patients / what is the rationale for the sample size? What is the hypothesis the authors would like to test? Additional info about these clinical trials shall be added as for any other clinical trials including patient selection criteria, rules to validate or reject a test, the test location, date, the detailed test protocol … the records should ideally be accessible…
Page 12 line 230: it is difficult to see a logarithmic function here
Author Response
Please see the attachment."

Round 2
Reviewer 1 Report
Authors have adequately addressed most of the comments raised
Author Response
Thank you very much again for your careful review. We are delighted that you have been satisfied with our response and revision.
Reviewer 2 Report
Dear authors, thank you very much for considering the comments made during the review process. Due to the limited time available to send a revised manuscript, I can understand that it is difficult to proceed to further tests. The new manuscript already showed significant improvements and may be considered for publication after considering one last minor change about the flow rate unit used in the new Table 1 and Table 2. I guess it is milliliter per minute or ml/min, it is preferable to not use ml/m because we can understand ml/meter.
Author Response
Thank you very much again for the second review. We have revised the paper to modify the unit to ml/min in Table 1 and 2.